# Mechanical-Resonance-Enhanced Thin-Film Magnetoelectric Heterostructures for Magnetometers, Mechanical Antennas, Tunable RF Inductors, and Filters

**DOI:** 10.3390/ma12142259

**Published:** 2019-07-13

**Authors:** Cheng Tu, Zhao-Qiang Chu, Benjamin Spetzler, Patrick Hayes, Cun-Zheng Dong, Xian-Feng Liang, Huai-Hao Chen, Yi-Fan He, Yu-Yi Wei, Ivan Lisenkov, Hwaider Lin, Yuan-Hua Lin, Jeffrey McCord, Franz Faupel, Eckhard Quandt, Nian-Xiang Sun

**Affiliations:** 1Department of Electrical and Computer Engineering, Northeastern University, Boston, MA 02115, USA; 2College of Engineering, Peking University, Beijing 100871, China; 3Institute for Materials Science, Kiel University, Kaiserstraße 2, 24143 Kiel, Germany; 4Winchester Technologies LLC, Burlington, MA 01803, USA; 5Materials Science and Engineering, Tsinghua University, Beijing 100084, China

**Keywords:** thin film, magnetoelectric (ME) heterostructures, multiferroic devices, magnetometers, mechanical antennas, tunable RF devices

## Abstract

The strong strain-mediated magnetoelectric (ME) coupling found in thin-film ME heterostructures has attracted an ever-increasing interest and enables realization of a great number of integrated multiferroic devices, such as magnetometers, mechanical antennas, RF tunable inductors and filters. This paper first reviews the thin-film characterization techniques for both piezoelectric and magnetostrictive thin films, which are crucial in determining the strength of the ME coupling. After that, the most recent progress on various integrated multiferroic devices based on thin-film ME heterostructures are presented. In particular, rapid development of thin-film ME magnetometers has been seen over the past few years. These ultra-sensitive magnetometers exhibit extremely low limit of detection (sub-pT/Hz^1/2^) for low-frequency AC magnetic fields, making them potential candidates for applications of medical diagnostics. Other devices reviewed in this paper include acoustically actuated nanomechanical ME antennas with miniaturized size by 1–2 orders compared to the conventional antenna; integrated RF tunable inductors with a wide operation frequency range; integrated RF tunable bandpass filter with dual H- and E-field tunability. All these integrated multiferroic devices are compact, lightweight, power-efficient, and potentially integrable with current complementary metal oxide semiconductor (CMOS) technology, showing great promise for applications in future biomedical, wireless communication, and reconfigurable electronic systems.

## 1. Introduction

Multiferroic materials, by definition, possess at least two of the ferroic properties (ferroelectricity, ferromagnetism, or ferroelasticity). In such materials, the interactions between the different order parameters can lead to new effects, such as magnetoelectric (ME) effects [1,2,3,4]. According to the physical control mechanism, ME effects are classified as two types: direct ME coupling and converse ME coupling. By definition, direct ME coupling refers to magnetic field control of electric polarization. In contrast, converse ME effect refers to electric field manipulation of magnetization. From the view of material constituents, the multiferroic ME materials can be categorized into two groups, namely single-phase and composite materials. The single-phase multiferroic materials are natural compounds, whereas the composites typically incorporate ferromagnetic and ferroelectric phases. The magnetoelectric responses of single-phase multiferroic materials, such as BiFeO_3_, are either weak or occur at temperatures too low for practical applications [5,6]. In contract, multiferroic composites typically exhibit very large ME coupling (several orders of magnitude higher than that of single-phase multiferroics) above room temperature [7]. It should be noted that neither the ferromagnetic nor ferroelectric phase has a ME effect, but the cross elastic interaction between the phases yields a remarkable ME effect. The strong ME coupling found in multiferroic composites proves that efficient energy conversion occurs between the electric and magnetic fields via elastic interaction, making multiferroic ME composites promising candidate to realize a wide variety of devices such as magnetometers [8,9], tunable radio frequency (RF) devices [10,11,12], transformers [13,14,15], gyrators [16,17,18,19], energy harvesters [20,21], spintronics [22,23], and random access memory [23,24,25]. In addition to strain-mediated ME coupling, other ME coupling mechanisms such as charge-mediated ME effect [26,27] and exchange bias-mediated ME effect [28,29] have also received a lot of attention. However, these ME coupling mechanisms are still in the early research stages and not discussed in this paper. Some comprehensive review papers for various ME coupling mechanisms are available [5,6,30].

Generally, there are two types of ME composites, namely bulk ME composites and thin-film ME heterostructures. During the last two decades, much effort has been focused on the theoretical and experimental investigations of bulk ME composites. Excellent reviews on the bulk ME composites can be found elsewhere [3,5,7]. The investigation of thin-film ME heterostructures has recently accelerated due to the advances in thin-film growth techniques [31]. Compared to the bulk composites, thin-film ME heterostructures benefit from small form factor, lightweight, low cost due to batch fabrication, as well as the potential capability to integrate with CMOS circuit [32]. In addition, thanks to the advancement of thin-film deposition techniques, excellent elastic interactions can be readily achieved between the piezoelectric and magnetostrictive films. The ME coupling between the two types of films via elastic interaction becomes maximum at the mechanical resonant frequency of the heterostructures, which can be essentially considered as mechanical resonators. Such resonance-enhanced ME coupling inherent to the thin-film ME heterostructures, together with the excellent elastic interactions between the constituent phases, greatly benefit the performance of the integrated devices based on thin-film ME heterostructures. These unique characteristics make thin-film ME heterostructure more advantageous compared to its bulk counterpart in certain applications where miniaturization of the device is crucial. One good example is the magnetometer array which usually requires high spatial resolution and, thus, small size for each constituent magnetometer.

In this article, we present the most recent progress on several types of integrated multiferroic devices based on thin-film ME heterostructures, such as magnetometers, mechanical antennas, tunable RF inductors, and filters. Table 1 provides the summary of the physical mechanisms for different types of integrated devices. Among these devices, the working principles of magnetometers and mechanical antennas in receiving mode are both based on direct ME coupling, which refers to the occurrence of electrical polarization when the devices are subjected to a magnetic field (or waves). In contrast, the working principles of RF tunable devices and mechanical antennas in transmitting mode are both based on converse ME coupling, which refers to the change of magnetic properties due to the application of an electric field. Interestingly, both direct and converse ME coupling are employed in the mechanical antennas, which enables the function of receiving and transmitting using the same thin-film ME heterostructure.

## 2. Thin-Film Characterization Techniques

The strength of ME coupling in the thin-film heterostructures is determined by many factors, such as the properties of the two constituent phases, the elastic interaction between them, the vibration mode of heterostructure, and the orientation of the magnetic and electric fields. Among these factors, the film properties of the two constituent phases are the most fundamental ones in determining the strength of ME coupling. For example, large ME coupling requires large a piezoelectric coefficient in the piezoelectric phase and a large piezomagnetic coefficient in the magnetostrictive phase due to the fact that ME coupling is a product tensor property which is a result of the product of the piezoelectric effect and magnetostrictive effect. In addition, film properties such as coupling coefficient, permittivity and dielectric loss of piezoelectric phase, magnetostriction, saturation field, anisotropic energy, coupling coefficient, permeability and loss tangents associated with the magnetostrictive phase also play significant roles in determining the performance of the thin-film ME heterostructures. Given that these film properties are varied a lot depending on the adopted fabrication processes and conditions, it is crucial to evaluate these properties using thin-film characterization techniques for the purpose of design, simulation, and performance optimization of the integrated multiferroic devices. Comprehensive summary of the film properties can be found for both piezoelectric [33,34] and magnetostrictive thin films [35,36].

### 2.1. Characterization of Piezoelectric Thin Films

Among all the piezoelectric thin films, lead zirconate titanate (PZT) and aluminium nitride (AlN) are considered as representative materials. PZT is a polycrystalline ceramic and belongs to ferroelectric materials. The advantage of PZT films is the high electromechanical coupling coefficient and piezoelectric coefficient. These superior properties are normally obtained at morphotropic phase boundary (MPB) composition, which corresponds to Zr/Ti ratios close to 58/42 [37]. The main drawback of PZT films is the presence of lead in this material, which causes higher probability of contamination during its processing and not suitable for biological applications. It is also reported that PZT films exhibit high acoustic loss at high frequency range [38]. In contrast to that, sputter-deposited AlN film, being textured polycrystalline non-ferroelectric material, is a better choice for high-frequency applications due to is low dissipation factor and moderate piezoelectric coefficient [39]. More importantly, AlN films are biocompatible and also compatible with the standard CMOS technology [40,41]. 

Extensive works have been carried out to develop the characterization techniques for PZT and AlN thin films [42,43,44,45,46]. For example, the crystallization quality in AlN film can be evaluated by the X-ray diffraction (XRD) technique. The scanning electron microscopy (SEM) and atomic force microscopy (AFM) techniques were commonly used to examine the topography and surface roughness of the films. The permittivity and dielectric loss of the films can be obtained by measurement using impedance analyzer. To measure the piezoelectric coefficient of the films, a direct measurement approach based on double-beam interferometer was often used, the main characteristic of which is to discriminate the true film expansion from the bimorph-like binding of the sample [42,47]. In addition, many indirect measurement techniques based on cantilever structures [37], surface-acoustic-wave (SAW) resonators [48], and bulk-acoustic-wave (BAW) resonators [38] were also demonstrated to be effective to extract the piezoelectric coefficients.

### 2.2. Characterization of Magnetostrictive Thin Films

The most commonly used magnetostrictive materials are magnetic alloys such as Terfenol-D, Galfenol, FeGaB, and FeCoSiB. Terfenol-D (Tb_0.3_Dy_0.7_Fe_1.92_), with a saturation magnetostriction constant up to 1600 ppm, is well-known as a giant magnetostrictive material [49]. Although having large saturation magnetostriction, Terfenol-D also exhibits a saturation field as large as several kOe, which makes its piezomagnetic coefficient small [50]. On the other hand, Galfenol (binary FeGa alloy) shows a large saturation magnetostriction constant of near 300 ppm with low saturation field in the order of 100 Oe, but the problem of Galfenol is its large loss tangents at RF/microwave frequency range, which was manifested as large ferromagnetic resonance (FMR) linewidth of 450–600 Oe at X band [51]. It has also been demonstrated that incorporation of metalloid element (i.e., boron) into magnetic binary alloys help refine the grain size and reduce magnetic anisotropy, resulting in excellent soft magnetic properties such as low saturation field, large saturation magnetostriction as well as low loss tangents [52]. Among the thin films based on soft magnetic alloys, FeCoSiB and FeGaB thin films are most representative and have been successfully demonstrated to build ME heterostructures with giant ME voltage coefficients and low loss tangents at microwave frequencies [53,54]. 

Many characterization techniques were developed to evaluate these metallic soft magnetic films. Compositions and crystal structures of the films can be examined by X-ray photoelectron spectroscopy (XPS) and XRD, respectively. Magnetic properties, such as coercive field, saturation field, and saturation magnetization, can be measured using magnetometry techniques based on vibrating sample magnetometers (VSM), superconducting quantum interference devices (SQUID), or BH-loopers. The permeability, loss tangents and FMR linewidth can be extracted from the measured S parameters of a thin-film loaded waveguide which is immersed in a fixed magnetic field [55]. As for measurement of magnetostriction, several techniques have been proposed, which can be broadly classified as either direct or indirect methods [56]. Direct methods enable the magnetostriction to be measured directly as a function of the applied magnetic field. The most commonly used direct methods include the capacitance method based on cantilever structures [57], the optical method utilizing rotating magnetic fields [58], and the nano-indentation method [59]. In the capacitance method, the cantilever with the thin film on top forms one plate of a capacitor, which serves as a tuning capacitance in an oscillator. The magnetostriction can be derived by measuring the shift in resonant frequency of the oscillator, which is caused by the capacitance change due to deflection of the cantilever. The drawback of the capacitance method is that its accuracy is easily impacted by the environment, such as ambient temperature. The optical method utilizing rotating magnetic fields permits measurement with high accuracy and does not need any contact between the probe and sample. However, the method requires a large space and a sophisticated optical system. The nano-indentation method enables measurement of both elastic properties and magnetostrictions in thin films, but it requires a minimum penetration depth into the film (60 nm) and causes unrecoverable damage to the film. In addition to magnetostriction, the change in elastic modulus as a function of applied magnetic field (delta-E effect) is also an important magnetomechanical property of thin films, given that increased interest has recently been spurred in employing the delta-E effect to build sensitive magnetometer [32,60,61]. Unfortunately, few works have been reported to quantitatively characterize the delta-E effect [62]. 

Recently, Dong et al. [63] developed a simple, compact, and sensitive system to measure magnetostriction, the piezomagnetic coefficient, and the delta-E effect by using a non-contact optical technique. The system includes a magnetostriction tester and a delta-E tester, both of which were based on a similar structure and can be assembled with very few parts. Figure 1a shows the schematic of the proposed magnetostriction tester in three-view. An adjustable AC driving magnetic field ranging from 0 to 300 Oe is provided by two pairs of mutually perpendicular set Helmholtz coils. Equivalent amplitude and 90° phase shift current in the two pairs of Helmholtz coils can be configured to generate a rotating magnetic field. The sample under test is prepared by depositing the magnetostrictive thin film on a cantilever. An MTI-2000 fiber-optic sensor is utilized to detect the deflection of the cantilever tip due to strain changes in the magnetostrictive thin film. The displacement between the cantilever tip and the optical probe can be detected and converted to an AC voltage signal, which is recorded using a SR-830 lock-in amplifier. As a result, the magnetostriction and piezomagnetic coefficient of the magnetostrictive thin-film can be deduced. It should be noted that the AC driving frequency should be set to off resonance to avoid mechanical amplification. The schematic of the proposed delta-E tester is shown in Figure 1b, where it can be seen that the measurement setup is very similar to that of magnetostriction tester. A 0.1 Oe AC driving field with the frequency swept around 1555 Hz (second resonant frequency of the cantilever) is provided by a solenoid, which will excite the cantilever to vibrate at its second resonant mode. The reason to choose the second resonant mode is that the frequency of the fundamental vibration mode is too low (280 Hz), and the resultant miniscule frequency shift is very difficult to distinguish. For the higher-order resonant modes, the signal strength becomes too weak to detect. Outside the solenoid, one pair of Helmholtz coils is set to provide the DC bias field, which will induce change in the magnetostrictive strain and, thus, leads to the change in Young’s modulus. By measuring the shift in resonant frequency, the change of Young’s modulus can be derived according to the modified Euler-Bernoulli beam theorem. To obtain the maximum magnetostrictive strain change and strongest vibration signal, the directions of the generated magnetic fields from the solenoid and Helmholtz coil should be set perpendicular to the easy axis of the thin films. 

To calibrate the system, some typical magnetostrictive thin films (e.g., Ni, Co, FeGa, and Permalloy) with reported magnetostriction should be used. Figure 1c shows the measured magnetostriction of some magnetostrictive thin films after system calibration. To confirm the accuracy of the system, magnetostriction characterizations were carried out for as-deposited and annealed FeGaB thin films, respectively. Figure 1d shows the measured magnetostriction and piezomagnetic coefficient for as-deposited and annealed FeGaB thin films. It can be seen that the saturation magnetostriction field (15 Oe) is smaller for annealed FeGaB film compared to that of as-deposited one (20 Oe). At the same time, magnetic annealing also enhances the piezomagnetic coefficient from 7.5 ppm/Oe to 12 ppm/Oe. These improvements show that the total anisotropy energy was reduced by magnetic annealing, which essentially released the residual stress in the films. By using a delta-E tester, one can measure the change in Young’s modulus, as shown in Figure 1e. It can be seen that a larger reduction in Young’s modulus (153 GPa) occurs for annealed FeGaB thin films compared to that of the as-deposited one (125 GPa). Figure 1f shows the derivative of Young’s modulus in regard to the magnetic field (dE/dH), where it can be seen that the annealed FeGaB film exhibits a larger peak value of dE/dH (43 GPa/Oe). These measured results agree with the theoretical predictions, which suggests accurate measurement of the magnetostriction, piezomagnetic coefficient, and delta-E effect can be achieved by using the proposed system.

### 2.3. Magnetic Domain Characterization of Magnetostrictive Thin Films

Magnetic domain activity plays an important role for the noise characteristic in ME-based devices. Moreover, due to stress-strain effects [64,65] in the ME structures, the actual magnetic material properties, especially the effective magnetic anisotropy, may become local in nature. Only a few methods allow for a spatially resolved characterization of the magnetization response down to the device level. Magnetic domain observations techniques applied for the characterization of the piezomagnetic phase in ME heterostructures include magnetic force microscopy (MFM) [66], scanning electron microscopy with polarization analysis (SEMPA) [67], and soft X-ray magnetic circular dichroism–photoemission electron microscopy (XMCD-PEEM) [68]. In particular, magnetic domain imaging by magneto-optical Kerr effect (MOKE) microscopy [69,70] offers direct access to the local magnetization behavior on the ME device level [66,71,72,73,74,75,76]. MOKE microscopy allows in situ and time-resolved characterization of the magnetic phase of ME heterostructures. As shown in Figure 2a, characterization from a local level up to the extended ME structures is achievable. This includes obtaining quantitative information on the magnetic domain distribution. Figure 2b shows a quantitative domain image from the edge of a ME structure [65]. The characteristic domain pattern is because of stress relaxation at the device edges. The visible partial folding of magnetization is due to minimization of the magnetostrictive self-energy. Magnetoelastic interactions lead to an effective inversion of the magnetic anisotropy axis with effects on magnetization reversal and magnetic noise behavior in ME structures. Analyzing the spatial magnetic response also permits directly correlating the magnetic domain characteristics and ME response [65,70]. It was shown that the ME amplitude and working point, beyond material properties, are strongly dependent on the exhibited domain configuration. 

MOKE microscopy was also employed for process control for advanced ME schemes. In situ MOKE imaging during magnetic field annealing was utilized for magnetic domain engineering to improve ME device performance in terms of reducing the magnetic noise level [76].

## 3. Magnetometers

Until now, the gold standard for weak magnetic field measurement has been to use SQUIDs with a limit of detection (LOD) of a few fT/Hz^1/2^ at 1 Hz [77]. However, SQUIDs are bulky and expensive because their optimal performance is usually achieved with the help of liquid helium cooling for near absolute-zero temperature. Thus, it is highly desirable to find alternative magnetometer techniques that work at room temperature while also having a low LOD value. Recently, much effort has been focused on building ultra-sensitive magnetometers using ME heterostructures, which have been demonstrated to exhibit LOD at a few pT/Hz^1/2^ at 1 Hz without the need for cooling [78]. Such ME magnetometers are usually categorized into two groups, namely, bulk laminates or thin-film composites. The most important merit shared by bulk and thin-film ME magnetometers is their enhanced ME voltage coefficients and sensitivity (about 1 ~ 2 orders enhancement in magnitude) when driven at their mechanical resonant frequencies. This enhancement is due to the fact that much less energy loss, and thus much higher energy transduction efficiency, occurs at mechanical resonant frequencies compared to out-of-resonance frequency range. Different from the bulk ones, the thin-film ME magnetometers have much smaller size, which is essential for achieving high spatial resolution by forming magnetometer arrays [79]. Furthermore, thin-film ME laminates benefit from perfect elastic coupling between the magnetostrictive and piezoelectric phases. Another advantage of thin-film ME magnetometer is its capability to attain magnetic field sensitivity in only one direction, which is enabled by the in-plane anisotropy of the magnetostrictive layer. This feature makes it possible to realize magnetometer capable of three-dimensional vector field measurement. Currently, there are mainly three types of thin-film ME magnetometers based on direct or indirect detection schemes, which are respectively described in following three subsections.

### 3.1. Magnetometers Using Direct Detection

Magnetometers using direct detection are usually used for detecting AC magnetic fields. Their working principle is based on electrical polarization upon an applied magnetic field (direct ME effect). The applied AC magnetic field induces alternative magnetostriction in the magnetostrictive phase, which translates to stress variation in the piezoelectric phase and, thus, results in a change in the electrical polarization for output readout. A commonly used figure-of-merit to describe how sensitive the output electrical polarization responds to the applied magnetic field is the ME voltage coefficient. In 2009, Zhao et al. [80] demonstrated a thin-film ME magnetometer consisting of sol-gel-derived PZT and magnetron-sputtered FeGa layers on a micromachined silicon cantilever. The reported thin-film ME magnetometer shows ME voltage coefficient of 1.81 V/cm·Oe at a mechanical resonant frequency of 333 Hz under a DC bias magnetic field of 90 Oe. The purpose of the DC bias field was to shift the operating point of the magnetostrictive layer to the inflection point, where the derivative of the magnetostriction with respect to the magnetic field and, thus, the ME voltage coefficient reaches its maximum. With the help of magnetic field annealing, Greve et al. [53] demonstrated an extremely high ME voltage coefficient of 737 V/cm·Oe at a mechanical resonant frequency of 753 Hz under a DC bias magnetic field of 6 Oe in a ME magnetometer consisting of FeCoSiB/AlN thin-film ME heterostructures. The reported high ME voltage coefficient (737 V/cm·Oe) was two orders of magnitude higher than that at out-of-resonance frequency range (3 V/cm·Oe). The observed enhancement in ME voltage coefficient agrees with the measured quality factor (Q) of 140 at the resonant frequency. The demonstrated thin-film ME magnetometer exhibit pronounced magnetic field sensitivity only in the long axis of the cantilever, which was perpendicular to the magnetic field applied during the annealing process. This unique characteristic serves the possibility to realize a vector field magnetometer. To further enhance the ME voltage coefficient, Kirchhof et al. [81] operated the thin-film ME magnetometer in vacuum (under pressure below 10^−5^ mbar), demonstrating a giant ME voltage coefficient of 20 kV/cm·Oe at resonant frequency of 152 Hz, which represents a five-fold increase compared to that at atmospheric pressure. It is also shown that operating the magnetometer in vacuum improves the LOD by 10-fold (from 3 nT/Hz^1/2^ to 300 pT/Hz^1/2^). These improvements in ME voltage coefficient and LOD were attributed to much lower air damping in vacuum, which is manifested as a remarkable increase in Q from 1400 in air to 5800 in vacuum.

Most of the thin-film ME magnetometers mentioned above have similar laminate structures, which is FeCoSiB layer on top of AlN layer. The reason why AlN layer has to be deposited prior to deposition of FeCoSiB layer is due to that the conventional reactive magnetron sputter deposition of a high-quality AlN layer requires a substrate heating temperature as high as 400 °C [82], which negatively impacts the amorphous nature of FeCoSiB. However, it is also reported in [83] that this deposition sequence does not yield the best soft magnetic properties for FeCoSiB layer because of the higher roughness of AlN layer compared to that of silicon substrate. In 2016, Yarar et al. [84] developed an inversed deposition sequence, which allows depositing amorphous FeCoSiB on the smooth silicon substrate using magnetron sputter deposition, followed by sputter deposition of a high-quality AlN layer without intentional substrate heating [85]. The schematic illustration of the investigated magnetometer is shown in Figure 3a. The measured ME voltage coefficient as a function of the applied AC magnetic field was shown in Figure 3b, where it can be seen that a sharp peak (giant ME voltage coefficient as high as 5 kV/cm·Oe) occurs at the mechanical resonant frequency of 867 Hz. This suggests the resonance behavior of the ME heterostructure greatly enhances the ME voltage coefficient compared to that at out-of-resonance frequency range. Figure 3c depicts the measured ME voltage noise density as the magnitude of the applied 867 Hz magnetic field decreases. It can be seen that the measured data start to scatter as the magnetic field becomes smaller. This is due to the fact that the output electric signal from the magnetometer becomes indistinguishable from the noise when the signal-to-noise ratio (SNR) drops to 1. Thus, the value of LOD can be deduced from the readings marked by the red arrow, which is determined to be 400 fT/Hz^1/2^. Figure 3d shows the measured LOD at different frequencies. At low frequency range from 1 to 100 Hz where noise with 1/f characteristic dominates, measured values of LOD are in the nT/Hz^1/2^ range. A remarkable improvement in LOD (below 1 pT/Hz^1/2^) can be seen around the mechanical resonance. As the frequency is further increased, the noise restricts the LOD within a few pT/Hz^1/2^ range.

There is one potential problem for the cantilever-type ME magnetometer, which is that such a magnetometer is vulnerable to extrinsic acoustic and vibrational noise since these noises are usually in the same frequency range as the resonant frequencies of the bending modes. To reduce the noise contribution, Salzer et al. [86] proposed a noise cancellation scheme by placing two cantilever-type magnetometers in an inverse orientation (tuning fork). Such an orientation setup allows two cantilevers bending in opposite directions producing a differential-mode signal when a magnetic field is applied, but bending in the same direction resulting in a common-mode signal upon vibrational excitation. Thus, acoustic and vibrational noise can be highly suppressed when simply taking the difference of the signal components utilizing an instrumental amplifier. Using such a noise cancellation method, LOD of approximately 500 fT/Hz^1/2^ has been achieved at a resonant frequency of 958 Hz. 

### 3.2. Magnetometers Based on Frequency Conversion

One problem for magnetometers using direct detection is narrow bandwidth. This problem occurs because the optimal ME voltage coefficient and LOD can only be obtained when the frequency of the magnetic signal to be detected matches the resonant frequency of the ME composite. For bio-magnetic applications such as measurement of heart signals, the frequency range of the magnetic signal to be detected is from 0.1 Hz to approximately 100 Hz. Mechanical resonators with such low resonant frequencies would have excessively large size, which makes it impractical to realize using thin-film ME heterostructure. Thus, most of the cantilever-type thin-film ME magnetometers have resonant frequencies from several hundred Hz to a few kHz. This means that the ME voltage coefficient will be much lower at the frequency range of interest below 100 Hz due to the missing resonance enhancement. Furthermore, LOD decreases due to dominating 1/f noise contribution towards the low frequency range. Recently, two indirect detection schemes have emerged showing promises to solve this problem: magnetometers using frequency conversion and magnetometers based on delta-E effect. The frequency conversion techniques allow transferring the desired low-frequency magnetic signal to the higher resonant frequency of the magnetometer structure. Currently, there are two available approaches to realize frequency conversion which are using magnetic frequency conversion or electric frequency conversion.

The frequency-conversion-approach for thin-film ME magnetometer was first demonstrated by Jahns et al. [87], which used a time-varying bias magnetic field (modulation signal) instead of a DC bias field commonly adopted in direct detection method. Due to the non-linear characteristics of the magnetostriction curve, the frequency of desired magnetic signal will be upconverted to the mechanical resonant frequency of the ME magnetometer under the premise that the amplitude of the desired signal is much smaller than that of the modulation signal. This technique offers the possibility to achieve resonance enhanced ME voltage coefficient and improved LOD at virtually arbitrary frequencies far below mechanical resonant frequency, which allows wideband measurement. It was shown in [87] that this frequency conversion technique using magnetic modulation improved LOD by three orders of magnitude (from 1 µT/Hz^1/2^ to 1 nT/Hz^1/2^) upon desired magnetic signal at 1 Hz. However, this value of LOD is still much larger than that using direct detection method (7.1 pT/Hz^1/2^ [87]). The reason for this is that the undesired occurrence of additional noise under modulation, which was systematically studied in [88].

To reduce the noise caused during magnetic frequency modulation, Röbisch et al. [75] reported an effective method, which utilize exchanged biased (EB) multilayers as a magnetostrictive phase. The introduction of the exchange bias coupling in magnetostrictive multilayers, magnetic domain configurations can be systematically altered, highly reducing the noise caused by magnetic frequency modulation. As a result, a LOD of 180 pT/Hz^1/2^ upon a desired magnetic signal at 10 Hz has been demonstrated, which represents an enhancement in the LOD by a factor of about five compared to the magnetometer without exchange bias layer. A further step in improving the noise characteristics of ME magnetometers has been taken by an almost complete elimination of magnetic domain activity using an antiparallel exchange bias scheme [76]. By this, the shape and demagnetization effects are overcome and stable single-domain configurations in the magnetic sensing layers of the employed magnetic multilayers are formed. As a result, magnetic noise contributions are eliminated and LODs down to 60 pT/Hz^1/2^ at a magnetic signal of 10 Hz are obtained [89]. The antiparallel bias arrangement can also be applied to other ME heterostructures. It offers the possibility of further improvement in LOD, envisioning LODs below 10 pT/Hz^1/2^. Another advantage of using exchange bias coupling refers to elimination of an external DC magnetic bias field, which favors reducing the complexity of the measurement setup.

One drawback for the magnetic modulation is the large power consumption for generating AC magnetic fields, thus prohibiting the construction of sensor arrays. To overcome this issue, Hayes et al. [90] reported a cantilever-type magnetometer using electric frequency conversion technique, which consumes much less power by employing electrical field to actively drive the ME heterostructure. Figure 4a shows the schematic of this device, where it can be seen that the ME heterostructure mainly consists of two electrically independent piezoelectric phases (AlN layer and PZT layer) and one magnetostrictive phase (EB-FeCoSiB layer). The electric modulation signal is applied across the unpoled PZT layer to excite the ME heterostructure to vibrate. This excitation method enables frequency conversion since PZT shows a nonlinear piezoelectric coefficient. The AlN layer is used for the detection of the output voltage. This readout scheme is considered to be more compact compared to the one using a pickup coil wound around the magnetometer [91]. Figure 4b shows the measured output spectrum taken from the AlN layer when a low frequency 2 Hz sinusoidal magnetic field with an amplitude of 1 µT is present. No DC magnetic bias field is required since the exchange biased magnetostrictive multilayer (EB-FeCoSiB layer) shows its maximum magnetoelectric response in zero field. It can be seen that the output signal reaches its maximum at 689 Hz, which agrees with the resonant frequency of the ME heterostructure. Two equally large side-band signals are formed at f_mod_ ± 2 Hz due to the frequency modulation. A limit of detection of approximately 5 nT/Hz^1/2^ can be derived upon the fact that a 1 µT signal is about 45 dB above the noise floor. Figure 4c shows the case when a 20 Hz sinusoidal magnetic field with an amplitude of 1 µT is applied. The frequency of the modulation signal applied to the PZT layer is set to be 669 Hz. Thus, the sum of the modulation frequency (669 Hz) and the desired magnetic signal (20 Hz) matches the resonant frequency of the ME composite (689 Hz). It can be seen that the up sideband signal at 689 Hz is much stronger than the low sideband signal due to the enhancement of the Q at the resonant frequency. Figure 4d shows the linearity measurement performed at magnetic field of 10 Hz, where it can be seen that SNR drops to 1 at 10 nT/Hz^1/2^. This translates to a LOD value of 10 nT/Hz^1/2^. 

In order to retain power consumption at a minimum and avoid external magnetic bias or driving fields, replacing one piezoelectric layer with a mechanically decoupled passive pickup coil is an option. The coil surrounds a similar thin film composite with which the converse magnetoelectric effect is exploited [91]. The setup is shown in Figure 5a. Exciting of a high frequency resonance mode at about 500 kHz, causes magnetoelastic interactions with the magnetostrictive phase, leading to periodic magnetic flux change within the coil. The coil voltage amplitude at this frequency shows a strong dependence towards external magnetic fields exceeding 2.2 kV/T, shown in Figure 5b. A linear resolution of 1.2 nT at quasi-DC of 200 mHz is achieved in this way, indicated by 15 dBV of noise margin in Figure 5c,d. Utilizing such high resonant frequency modes provides inherently superior acoustic noise rejection.

### 3.3. Magnetometers Based on Delta-E Effect

An alternative method for magnetic field sensing utilizes the change of the elastic constants of magnetostrictive materials with magnetization. It arises from an additional magnetostrictive strain that adds up to the Hookean strain upon application of a stress. This effect is commonly referred to as the delta-E effect and has been extensively studied experimentally and theoretically in the beginning of the last century [92,93,94,95,96,97]. Later studies were focused on the Young’s modulus and the shear modulus of soft magnetic amorphous ribbons and wires for magnetic field sensing [98,99,100,101,102], due to their soft magnetic properties and easy processing. Although, shear wave magnetometers demonstrated the detection of low frequency magnetic fields in the 100 pT regime [103], the usage of macroscopic amorphous ribbons limits the device integration. First integrable delta-E effect sensors [104] were improved using magnetoelectric composites [60,61] with soft-magnetic amorphous thin films [63,102]. Via the piezoelectric layer the resonator is excited to oscillate at its resonant frequency and read-out simultaneously. Upon application of a magnetic AC field, the resonant frequency changes, which modulates the read-out current in amplitude and in phase [105] and mixes the signal up to higher frequencies. This permits delta-E effect magnetometers to detect DC or low-frequency magnetic signals in a comparatively large bandwidth, avoiding low frequency 1/f noise (f: frequency) and acoustic disturbances. Different resonator designs have been demonstrated. With cantilever resonators, equivalent magnetic noise levels of about 100 pT/Hz^1/2^ between 10 and 100 Hz were reached [106] using the second transversal bending mode. Such devices benefit from magnetically well-oriented FeCoSiB-Cr multilayers and an electrode design adapted to the resonant mode of the cantilever, as shown in Figure 6a. To obtain the largest piezoelectric transduction for the second transversal bending mode as shown in Figure 6b, the location of the two electrodes (i.e., E_1_ and E_2_) was designed to match with the positions where maximum stress occurs along the cantilever.

Simultaneous excitation of different resonant modes was investigated for noise cancelling and detection limit improvement [107] under ambient conditions. Experimental and theoretical studies focused on the origin of signal and noise levels, whereas, at small carrier amplitudes and small signal frequencies, the thermal-mechanical noise [108] dominates the noise floor, the overall detection limit is currently restricted by the onset of magnetic noise at large excitation amplitudes [109,110]. In contrast to the direct ME operation, the quality factor of the resonator has a more significant influence on the detection limit, which is found to be proportional to Q^−1.5^ [110].

For array applications and a high local magnetic field resolution, smaller sensors are more advantageous. Nan et al. [61] demonstrated a thin-film ME magnetometers using a nano-plate structure with a resonant frequency of 215 MHz. This high resonant frequency is achieved using interdigital electrodes to launch Lamb waves in the nano-plate. The working principle of Lamb wave resonators can be found in [111,112]. As shown in Figure 7a, the nano-plate structure consists of a (FeGaB/Al_2_O_3_) × 10 multilayer with total thickness of 250 nm, serving as magnetostrictive phase as well as a floating top electrode, and a 250 nm-thick AlN layer serving as piezoelectric phase. Patterned Pt layers are used as interdigital electrodes at the bottom of the AlN layer. The reason to choose an FeGaB/Al_2_O_3_ multilayer as magnetostrictive phase is because it exhibits a high piezomagnetic coefficient and low eddy current loss at radio frequencies [52,113]. When a magnetic field is applied, the Young’s modulus of the FeGaB layer changes due to delta-E effect, thus resulting in a change in the resonant frequency of the nano-plate. An SEM image of the fabricated nano-plate magnetometer is shown in Figure 7b. Figure 7c shows the measured magnitude of the admittance for the magnetometer under different DC magnetic bias fields applied along the length direction of the device. The dependency of the resonant frequency and peak admittance on the DC bias magnetic field is plotted in in Figure 7d. It can be seen that both resonant frequency and peak admittance decrease with DC bias magnetic field at low bias level, reaching minimum values at 15 Oe bias field, and then increase until the magnetostrictive phase (FeGaB/Al_2_O_3_) is saturated at about 50 Oe. The observed enhancement of Q at large magnetic fields is due to the elimination of magnetic domain walls and reduction of magnetic losses. As shown in Figure 7e, the magnitude of the peak admittance was recorded as the magnetic field decreases from 200 nT to 0.1 nT in a magnetically unshielded environment. The magnitude of admittance starts to scatter at 300 pT, indicating that the signal to be detected cannot be distinguished from noise below 300 pT. This translates to minimum detectable DC magnetic fields of 300 pT. Moreover, a minimum detectable DC magnetic field of 600 pT is achieved under self-bias condition, which is shown in Figure 7f.

A completely different approach of using the delta-E effect for magnetic field sensing employs surface acoustic waves (SAW) devices [114]. In order to enhance the effect, Love wave SAW devices are used. Here, shear horizontal acoustic surface waves are concentrated at the surface of the device by an amorphous SiO_2_ guiding layer which is coated with a thin amorphous magnetostrictive FeCoSiB film exhibiting the pronounced delta-E effect [115]. By this concentration of the acoustic waves at the surface, the influence of the delta-E effect on the wave propagation is maximized. This SAW magnetic field sensor can be operated in a delay line configuration and translates the magnetic field-induced changes of the shear modulus via the corresponding velocity changes of the Love waves to a time delay and a related phase shift. Another alternative is to measure the related frequency changes in a resonant configuration [116]. Figure 8a shows the sensor design of the Love wave surface acoustic wave sensor based on ST-cut quartz substrate and a 4.5 μm SiO_2_ guiding layer. The 200 nm magnetostrictive FeCoSiB thin film is deposited on the delay line between the interdigital transducers, which have a periodicity of 28 μm. Important characteristics include a low noise level of 250 pT/Hz^1/2^ at 10 Hz as shown in Figure 8b, a dynamic range covering six orders of magnitude and a possible bandwidth exceeding 1 MHz.

### 3.4. Practical Applications of Thin-Film ME Magnetometers

The thin-film ME magnetometers have shown LOD of 1 pT/Hz^1/2^ or even below in mechanical resonant frequencies [75,86]. Given the reported low values of LOD, a hypothesis that thin-film magnetometer could be used for biomagnetic measurements was often mentioned but never proven. The first proof of this hypothesis was recently reported by Reermann et al. [117], where the first system using thin-film ME magnetometer for cardiological application was presented. Cantilever-type thin-film magnetometer was adopted with magnetic frequency modulation technique. In the reported system, sophisticated averaging techniques were applied to improve SNR and thus closed the gap between the measured LOD of the magnetometer and the magnetic signal from human heart. Moreover, to deal with various types of noise, different weighted averaging algorithms have been demonstrated to ensure convergence even in highly distorted environments.

Another practical application of thin-film ME magnetometers refers to magnetic particle mapping, which has been recently demonstrated by Friedrich et al. [118]. Imaging magnetic nanoparticles (MNPs) has gained widespread interest due to its great value in medical life science and diagnostic applications. The work presented in [118] shows that thin-film ME magnetometers can be used to localize MNPs, which is proven as a more cost-effective and efficient way compared to other available technologies. In the reported work, the nonlinear nature of the MNPs was utilized. When MNPs were exposed to a AC magnetic field that had 1/3 of the resonant frequency of the magnetometer, the third harmonic excitation due to the nonlinear behavior of the MNPs was measured by magnetometer with resonance enhanced performance, thus decoupling the desired and excitation magnetic fields. Furthermore, the reported work measured the magnetic response from 2D MNP distributions and reconstructed the distribution by solving the inverse problem. The measurement results suggests that the thin-film magnetometers hold great potential for mapping MNPs.

## 4. Mechanical Antenna

Conventional antennas using electric currents for electromagnetic wave radiation are ubiquitous in our daily life, which enables wireless communications in portable electronic devices such as smart phones, tablets, and radio frequency identification systems. A key challenge of conventional antennas is the size miniaturization since the size of the antenna should be designed to be comparable to the electromagnetic (EM) wavelength for optimal radiation efficiency. This miniaturization limitation causes difficulty in achieving compact antennas and antenna arrays at the very-high-frequency band (VHF, 30–300 MHz) and ultra-high frequency band (UHF, 0.3–3 GHz). Thus, new antenna concepts and radiation mechanisms for miniaturization of antennas are highly desired. To meet this demand, a novel antenna concept based on mechanically vibrating ME heterostructures were theoretically proposed recently in [119,120]. In 2017, Nan et al. [121] demonstrated the ultra-compact mechanical antennas based on suspended nano-mechanical ME resonators, which exhibit 1 ~ 2 orders of magnitude miniaturization over state-of-the-art compact antennas without performance degradation. The remarkable reduction in size benefits from the fact that the operation of mechanical antenna hinges on utilization of acoustic wave of the wavelength of which is about five orders of magnitude smaller than that of the electromagnetic wave at the same frequency. In [121], two types of structures for mechanical antennas based on laterally-vibrating or vertically-vibrating nano-mechanical resonators were proposed, which are described in the following subsections. 

### 4.1. Mechanical Antenna Based on a Laterally-Vibrating Nano-Plate Resonator

Figure 9a shows the SEM image of the laterally-vibrating nano-plate resonator (NPR) with a length of 200 µm and a width of 50 µm. The rectangular resonator body consists of a 50-nm-thick bottom Pt electrode and a thin-film ME heterostructure ((45 nm FeGaB/ 5 nm Al_2_O_3_) × 10/500 nm AlN) on top of the Pt layer. Another 100-nm-thick Au layer was patterned to form the ground-signal-ground (GSG) pads and top ground for electrical characterization. The multilayer nano-plate structure is fully released from the silicon substrate to minimize the substrate clamping effect, but mechanically supported by the anchors at two sides. The resonant frequency of the in-plane mechanical vibration mode of NPR is determined by the width of the structure, which offers designers great flexibility to choose the desired operation frequency. The working principle of such type of laterally-vibrating mechanical resonators is detailed in [122]. Figure 9b shows the measurement setup schematic with the structure of NPR. It can be seen that an RF coil, which was placed 14 mm away from the device under test, was used to generate an AC magnetic field at the frequency range around the mechanical resonant frequency of the NPR. Due to the direct ME effect, the AC magnetic field will induce strain variation in the magnetostrictive layer (FeGaB layer) which, in turn, transfers to the piezoelectric layer (AlN layer) and results in voltage output. The induced voltage output from the parallel-plate structure (FeGaB and Pt) can be detected by the lock-in amplifier. Figure 9c shows the measured amplitude of admittance of the NPR characterized by a network analyzer. The measured admittance spectrum was fitted into the Modified Butterworth-van Dyke model [123], from which the mechanical resonant frequency of 60.7 MHz, electromechanical coupling coefficient of 1.35% and quality factor of 930 were extracted. To confirm that the induced voltage output is due to the strain-mediated ME coupling, a nonmagnetic NPR using Cu as the top electrode instead of FeGaB was also measured using the same measurement setup. The measured result is shown in Figure 9d, where it can be seen that the nonmagnetic NPR exhibits a similar admittance spectrum compared to that of the NPR with magnetic film. Using the measurement setup illustrated in Figure 4b, the induced voltage output from the NPR under the RF coil excitation of 60 nT at 60.7 MHz was measured using a UHF lock-in amplifier. The measured result was shown in Figure 9e, where it can be seen that a peak voltage value of 180 µV occurs at the excitation frequency. This peak output voltage translates to ME voltage coefficient of 6 kV/cm·Oe, which suggests a strong ME coupling in the thin-film heterostructure. Figure 9f shows the measured output voltage from the Cu/AlN NPR under the same AC magnetic field excitation. It can be observed that the induced output voltage is about two orders of magnitude smaller than that in NPR with magnetic film. It is believed that this weak resonant signal is due to the inductive coupling between the RF coil and the device ground loop.

### 4.2. Mechanical Antenna Based on a Vertically-Vibrating Thin-Film Bulk Acoustic Wave Resonator

Acoustically-actuated antennas can also be designed to operate at the GHz range utilizing the thickness vibration mode of the ME heterostructure, which can be essentially seen as a thin-film bulk acoustic wave resonator (FBAR). The working principle of FBAR can be found in [124]. The same as other FBAR devices, the resonant frequency of the FBAR-type mechanical antenna is also set by the thickness of the ME heterostructure. Figure 10a shows the SEM of the vertically-vibrating FABR-type mechanical antenna, which has identical vertical laminate structure as NPR described in the previous section. The fully-released FBAR-type mechanical antenna has a lateral shape of a circular disk with a diameter of 200 µm. Figure 10b shows the schematic of the gain measurement setup for FABR-type mechanical antenna. It can be seen that this setup includes a calibrated linear polarization standard horn antenna and an FABR-type mechanical antenna under test. A network analyzer was used to electrically characterize the antenna by measuring its two-port S parameters. It is noted that this antennas measurement was performed in an anechoic chamber under a far-field configuration. In the receiving process, the magnetostrictive layer of the mechanical antenna detects the RF magnetic field component of electromagnetic waves, which induces oscillating strains in magnetostrictive layer. The strains are transferred to piezoelectric layer, and lead to AC voltage output, the magnitude of which reaches its maximum at mechanical resonant frequency. Conversely, during the transmitting process, the oscillating strains occur in the piezoelectric layer of mechanical antenna under input AC voltage. The strains are transferred to magnetostrictive layer, and excite magnetization oscillation (or magnetic current), which results in radiation. The measured magnitude of reflection coefficient (S_22_) of the FABR-type mechanical antenna is shown in Figure 10c, where the mechanical resonant frequency (2.53 GHz) can be found from the peak return loss of 10.3 dB. The inset of Figure 10c shows a simulated displacement profile of the device, which indicates a thickness vibration mode at 2.53 GHz. For comparison, a nonmagnetic control device using Al as the top electrode instead of FeGaB was also measured using the same measurement setup. The measured result was shown in Figure 10d, where a mechanical resonance frequency at 2.5 GHz can be found. Figure 10e shows a measured magnitude of transmission coefficients S_21_ and S_12_, which represent the transmitting and receiving behaviors of the mechanical antenna, respectively. The antenna gain is measured to be −18 dBi at resonant frequency through gain comparison method. In contrast, as shown in Figure 10f, only a very weak peak at 2.5 GHz can be observed for the control device which is about 20 dB lower than the performance of FABR-type mechanical antenna. This suggests that the ME coupling effect dominates the performance of the antenna transmission.

In summary, the ultra-compact mechanical antennas based on NPR or FBAR structures have antenna sizes one to two orders of magnitude smaller than the state-of-the-art compact antennas, thus providing great opportunities in realizing miniaturized antennas in future wireless communication systems such as Internet of Things, wearable monitoring systems, bio-implantable communication systems. Moreover, the mechanical antennas based on NPR or FBAR structures share the same five-mask microfabrication process on the same chip, which can be readily connected to CMOS oscillator circuits realizing reconfigurable mechanical antenna array with an operation frequency range from tens of MHz to tens of GHz.

## 5. Tunable RF/Microwave Devices

One important application of multiferroic heterostructure is to build electrostatically tunable RF/microwave devices, such as tunable inductors and filters. The function of tunability can be realized either by applying a magnetic field [125,126,127] or electrical field [128,129,130]. Compared to the conventional magnetically tunable devices, the electrostatically tunable multiferroic devices are more compact, lightweight, and power efficient. The working principle of electrostatically tunable multiferroic devices relies on the converse ME effect, which relates to E-field control of the magnetic permeability. Thus, it is of paramount importance to realize strong ME coupling in multiferroic heterostructure, which enables large tunability. However, it has been an open challenge to obtain large ME coupling at RF/microwave frequencies due to large loss tangents of magnetic phase. Moreover, the ideal magnetic phase should have large piezomagnetic coefficient, which translates to large saturation magnetostriction (λ_s_) and low saturation magnetic field (H_s_). Much effort has been focused on developing such high-quality magnetic thin films and building multiferroic heterostructures with large ME coupling at RF/microwave frequencies. Lou et al. [52] reported a class of metallic magnetic FeGaB films that has a high λ_s_ of 70 ppm, a low H_s_ of 20 Oe, and a narrow ferromagnetic resonance linewidth of 16 Oe at 9.6 GHz. The maximum piezomagnetic coefficient is about 7 ppm/Oe, which is much larger than that of other well-known magnetostrictive materials, such as Terfenol-D (Tb-Dy-Fe) [131], Galfenol (Fe-Ga) [132], and Metglas (FeBSiC) [133]. By depositing FeGaB film onto lead zinc niobate-lead titanate (PZN-PT) single-crystal slabs using magnetron sputter deposition, Lou et al. [54] demonstrated a novel FeGaB/PZN-PT heterostructure that exhibit high electrostatically tunable FMR frequency range of 5.82 GHz and a giant electrostatically induced magnetic anisotropy field of 750 Oe at both DC and microwave frequencies. Xing et al. [113] studied the structure of FeCoB/Al_2_O_3_ multilayer and found significant reduced eddy current loss as well as lower coercivity.

### 5.1. Tunable RF/Microwave Inductors

As a fundamental type of electronic component, inductors have been used ubiquitously in all kinds of electronic circuits. Compared to voltage tunable capacitors, which have been widely applied in a variety of electronic circuits, voltage tunable inductors are not readily available due to low inductance areal density and low quality factor especially at RF/microwave frequencies. Many integrated tunable inductors using MEMS technology were reported, which typically exhibit high quality factors [134,135,136]. However, the problem for these inductors is their low inductance areal densities due to lack of magnetic material. Salvia et al. [137] demonstrated an on-chip tunable inductor using patterned permalloy film. Measured results show a 40% increase in inductance by introducing the permalloy film, a 15% tuning range, and a Q between 5 and 11 up to 5 GHz. The drawback of this tunable inductor is its large power consumption which stems from the large DC current required for generating the magnetic bias field. A more power efficient voltage-tunable inductor using multiferroic heterostructure was proposed by Lou et al. [130]. The reported device was based on a solenoid inductor with multiferroic core which exhibits a large inductance tuning range of 450% with negligible power consumption.

Gao et al. [138,139] demonstrated a voltage-tunable RF integrated inductor based on solenoid structures using FeGaB/Al_2_O_3_ multilayer films. Compared with a single FeGaB layer, FeGaB/Al_2_O_3_ multilayer film with the same total thickness provides less eddy current loss, lower out of plane anisotropy, and higher permeability. Thus, the reported tunable inductor exhibits significantly enhanced inductance (>100%) and higher quality factor at GHz frequencies over their air core counterparts. The optical image of the tunable inductor is shown in Figure 11a, where a solenoid structure can be seen. The tunable inductor was fabricated using micro-fabrication techniques [138] followed by bonding to lead magnesium niobate-lead titanate (PMN-PT) piezoelectric slab. As shown in the Figure 11b, the magnetic phase (FeGaB/Al_2_O_3_ multilayer films) and piezoelectric phase (PMN-PT slab) form the multiferroic heterostructure. The function of voltage tuning in inductance is due to the strain-mediated ME coupling within the multiferroic heterostructure, which enables E-field-induced changes of the permeability of the FeGaB/Al_2_O_3_ multilayer film. Using a vector network analyzer together with a RF probing system, the inductance and quality factor of the tunable inductor can be measured. Figure 11c shows the measured inductances of the inductor with magnetic multilayer films as the inductor core and the one with air core. It can be seen that introducing the magnetic multilayer as the core doubles the inductance at frequency below 2 GHz. Figure 11d shows the measured inductance under different E-field applied across the thickness of the PMN-PT slab. The applied voltage ranges from 0 to 400 V which corresponds to an E-field from 0 to 8 kV/cm. It can be seen from Figure 11d that a high tunable inductance of more than 100% is achieved over a large frequency range from 2 GHz to 4 GHz. It was also reported that the measured Q of the inductor enhances by at least 50% over the frequency range from 0.5 GHz to 1 GHz.

### 5.2. Tunable RF/Microwave Filters

Filters are essential components in modern electronic systems since they provide the function of frequency selection. Wideband and low-loss tunable filters are highly desirable since they can greatly reduce the complexity in a wide variety of electronic systems such as reconfigurable electronic systems, multiband communication systems and wideband tracking receivers. One of the most practical implementation for RF/microwave tunable filters is based on yttrium-iron-garnet (YIG) resonators, which offers multi-octave bandwidths and high quality factor [140,141,142,143]. However, tunable filters based on YIG resonators are bulky and cannot be easily miniaturized for portable electronic systems. Moreover, the operation of YIG filters consumes considerable amount of DC power for generation of magnetic bias field [144]. Solid-state varactors have also been used to build tunable filters with wide tuning range, but they exhibit high loss and low linearity at RF/microwave frequencies [145,146,147]. In addition, it has been demonstrated that tunable filters based on MEMS switches and varactors provide low loss, low DC power consumption, and good linearity [148,149,150]. Another competitive alternative to realize a RF/microwave tunable filter is to utilize multiferroic heterostructures, which have been demonstrated to be compact, low-cost, and power-efficient. In particular, such a type of tunable filter provides the function of dual H- and E-field tunability, which offers more design flexibility.

Lin et al. [151] reported the first integrated dual H- and E-field tunable bandpass filters with nonreciprocal transmission characteristics. Figure 12a shows the schematic of the device, which mainly consists of an inverted-L-shape microstrip structure loaded with NiZn ferrite film, and a PMN-PT piezoelectric slab. The NiZn ferrite film is deposited using a spin spraying process, which is a low-temperature, low cost, and fast deposition method. The silicon substrate at the back side of the microstrip structure is removed using the deep reactive ion etching (DRIE) technique. A multiferroic heterostructure is formed by bonding the microstrip structure to the PMN-PT piezoelectric slab. By applying a magnetic bias field, the magnetostatic surface wave will be excited in the NiZn film [152]. The NiZn ferrite film is rotated by a certain angle and the reflection will propagate parallel to the bias field, which is operating in the stopband of magnetostatic back volume wave to diminish the splitting modes and the standing-wave resonances. The nonreciprocity characteristic is achieved from the limited surface wave on the bottom surface due to the non-reflection edges. Figure 12b,c show the measured S_11_ parameters of the tunable bandpass filter under DC magnetic field and DC electric field, respectively. It can be seen from Figure 12b that a tunable frequency range from 3.78 GHz to 5.27 GHz is achieved by increasing the DC magnetic field from 100 Oe to 400 Oe, which translates to an H-field frequency tunability of about 5 MHz/Oe. As shown in Figure 12c, the center frequency changes from 2.075 GHz to 2.295 GHz when the applied electric field across the thickness of PMN-PT increases from 0 kV/cm to 4 kV/cm. This represents a E-field frequency tunability of 55 MHz/(kV/cm). Figure 12d shows the measured S_21_ and S_12_ for the device under DC magnetic bias field of 400 Oe. It can be seen that an isolation of 15.8 dB is obtained between S_21_ and S_12_ showing nonreciprocal transmission characteristics.

More recently, an integrated RF tunable bandpass filter based on two coupled elliptic-shape nano-mechanical resonators was reported with dual E- and H-field tunability [153]. Figure 13a shows the schematic of the device, where it can be seen that each elliptic-shape resonator consists of a multiferroic heterostructure of magnetostrictive multilayer films (FeGaB/Al_2_O_3_) and piezoelectric film (AlN), together with a Pt layer as the bottom electrode. Two resonators are placed in close proximity with 2 µm air gap. Both resonators are excited in in-plane contour mode, the resonant frequency of which is determined by the lateral dimension of the resonator. Due to the strong ME effect between the FeGaB/Al_2_O_3_ multilayer and AlN film, the acoustic wave within each resonator can be strongly coupled via the electromagnetic waves. Figure 13b shows the measured S parameters for the filter under zero-bias field, where it can be seen that return loss of 11.2 dB and insertion loss of 3.4 dB are achieved at center frequency of 93.165 MHz with an extracted quality factor of 252. When a magnetic field is applied, the center frequency will change due to the magnetic field-induced Young’s modulus change in FeGaB film (delta-E effect). Figure 13c shows the measured resonant frequency as a function of applied DC magnetic field, where the H-field frequency tunability of 5 kHz/Oe can be extracted. The measured E-field frequency tunability is 2.3 kHz/V as shown in Figure 13d. It can be observed that the change in resonant frequency due to ME coupling is much smaller compared to that caused by the delta-E effect. This is due to the remarkable change in the Young’s modulus (up to 70%) due to the delta-E effect found in FeGaB film [63].

## 6. Conclusions

During the last decade, considerable progress has been made for different types of integrated multiferroic devices that are based on thin-film ME heterostructures. The performance of these integrated devices greatly benefit from the strong strain-mediated ME coupling occurred at the mechanical resonance of the magnetic/piezoelectric heterostructure. Despite the advantages of small form factor, low power consumption, and the feasibility of batch fabrication, excellent elastic coupling between the magnetic and ferroelectric phases can be attained thanks to the advancement of thin-film deposition techniques. It can be foreseen that further improvement on the magnetic and piezoelectric properties of thin-film materials will greatly benefit the performance of the integrated multiferroic devices. On the other hand, the accurate determination of these film properties also relies on the development of the thin-film characterization techniques.

A summary of characteristics for some of the reviewed integrated multiferroic devices is provided in Table 2. Among the presented devices, thin-film ME magnetometers have seen fast development in recent years. The achievable LOD of the magnetometer has been significantly improved by many novel techniques, such as magnetic/electric frequency conversion, utilization of delta-E effect in the magnetic phase, noise suppression method of exploiting tuning-fork structure and introduction of exchange biased magnetostrictive multilayers. For both DC and AC magnetic field detection, the best reported values of LOD achieved have been lowered into the range of pT/Hz^1/2^, making thin-film ME magnetometer a promising candidate for future biomedical and biomagnetic applications. In particular, some practical applications of using thin-film ME magnetometer for cardiological application and magnetic nanoparticles mapping were recently reported. 

A novel antenna miniaturization approach has been recently demonstrated in the mechanical antennas which are based on thin-film ME heterostructures. In this new antenna concept, transmitting and receiving behavior are realized by converse and direct ME coupling effects, respectively. The operation frequency of such mechanical antenna ranges from several MHz to tens of GHz, which can be well-controlled by the physical geometries of the thin-film ME heterostructures with in-plane or thickness vibration modes. Although still in its early stage, the mechanical antennas based on thin-film ME heterostructures have already exhibited one to two orders of reduced size compared to conventional antenna and ground plane immunity from metallic surfaces. One problem of such mechanical antennas is that the antenna gain is still not ideal, which could be solved by further optimization of the ME heterostructures or implementation of mechanical antenna arrays. With the advantages of high magnetic field sensitivity in near field, the reported mechanical antennas hold great potential in the future systems of biomedical communication, wearable electronics, and the Internet-of-Things.

Based on the converse ME coupling effects, novel integrated tunable RF inductors and nonreciprocal RF bandpass filters have been demonstrated. These integrated RF devices are compact, lightweight, power efficient, and have the potential to integrate with CMOS technology, providing great opportunities for future reconfigurable electronic systems.

## Figures and Tables

**Figure 1 materials-12-02259-f001:**
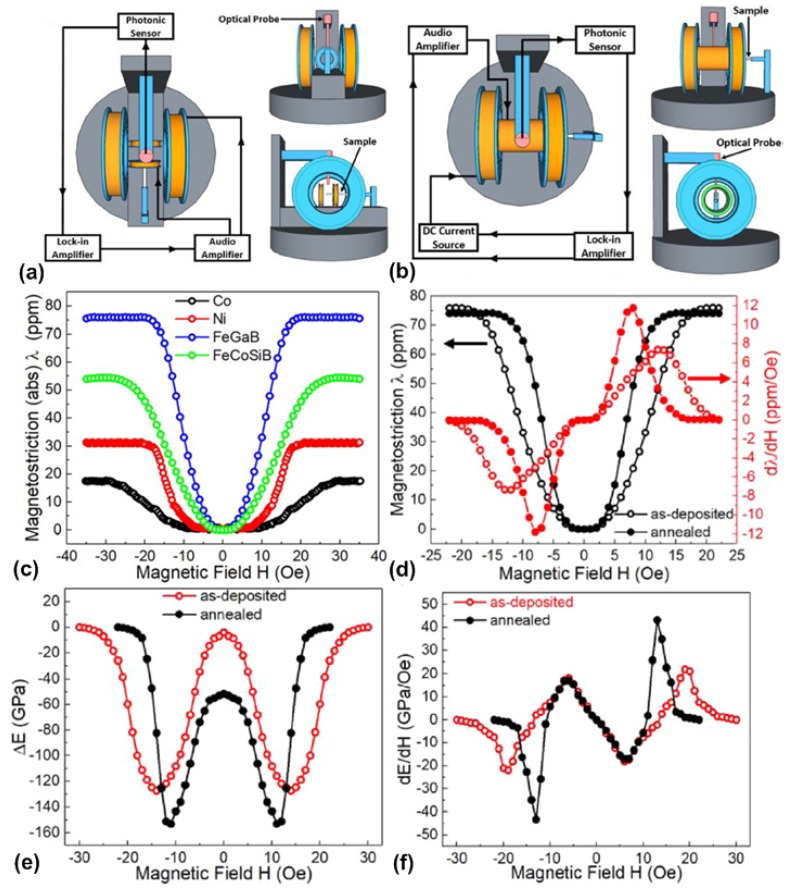
(**a**) Schematic of the proposed magnetostriction tester in three-view; (**b**) schematic of the proposed delta-E tester in three-view; (**c**) measured magnetostriction with respect to magnetic field for different magnetic thin films; (**d**) measured magnetostriction and piezomagnetic coefficient for as-deposited and annealed FeGaB thin films; (**e**) measured delta-E effect for as-deposited and annealed FeGaB thin films; and (**f**) measured dE / dH for as-deposited and annealed FeGaB thin films. This figure is reproduced with permission from American Institute of Physics (AIP) Publishing, [63].

**Figure 2 materials-12-02259-f002:**
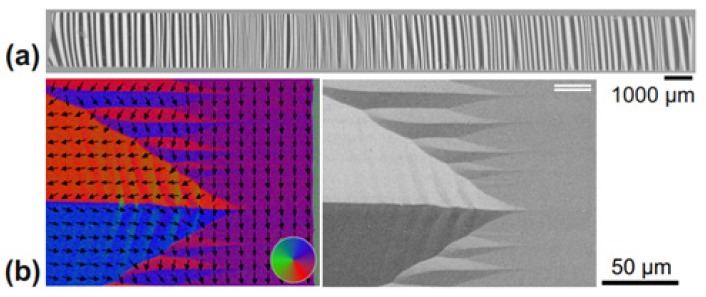
(**a**) Full magnetic domain distribution [70]; and (**b**) the quantitative magnetic domain evaluation and MOKE micrograph of edge closure domains states in a cantilever ME magnetometer based on FeCoSiB [67]. This figure is reproduced with permission from AIP Publishing, [70] and [67].

**Figure 3 materials-12-02259-f003:**
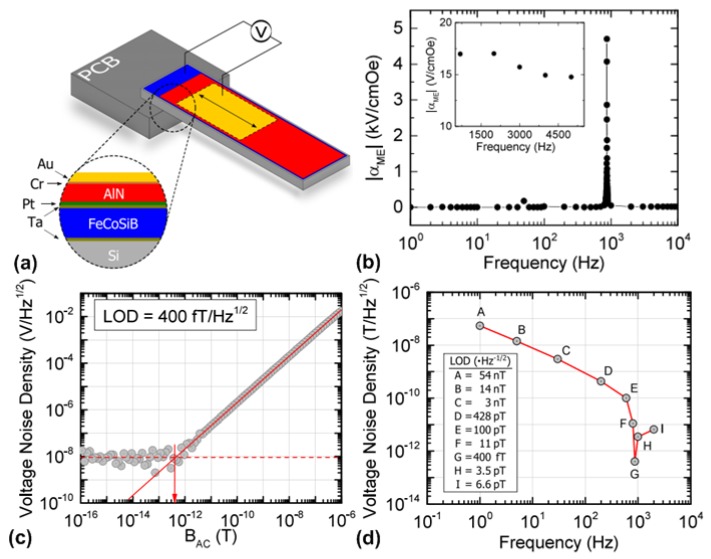
(**a**) Schematic of the cantilever-type magnetometer using inversed deposition sequence; (**b**) measured ME voltage coefficient as a function of the frequency of the applied AC magnetic field. The sharp peak marks the mechanical resonant frequency of 867 Hz. The inset shows the ME voltage coefficient above the resonance; (**c**) measured voltage noise density with respect to the magnitude of applied AC magnetic field with constant frequency of 867 Hz; and (**d**) measured voltage noise density with respect to the frequency of applied AC magnetic field with constant magnitude of 1 × 10^−3^ Oe. This figure is reproduced with permission from AIP Publishing, [84].

**Figure 4 materials-12-02259-f004:**
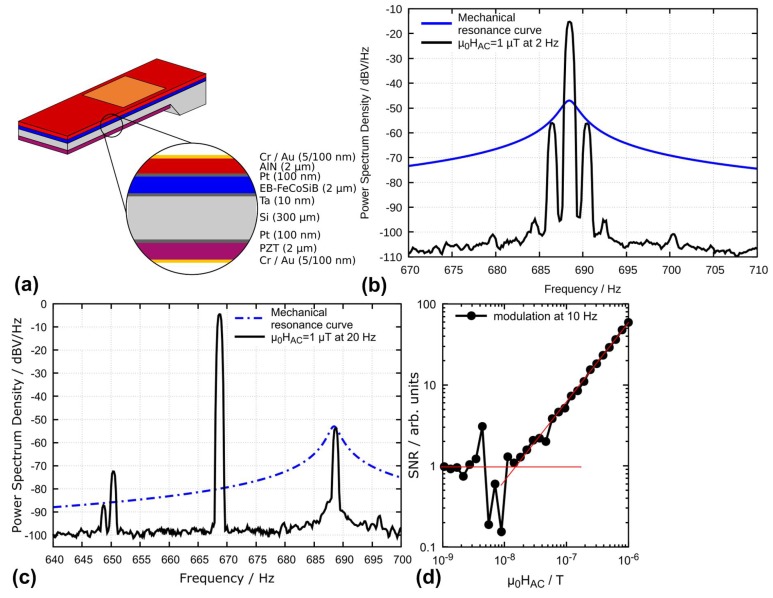
(**a**) Schematic of the cantilever-type magnetometer using electric frequency conversion; (**b**) measured output spectrum taken from the AlN layer when a low frequency 2 Hz sinusoidal magnetic field is present. The sensor’s PZT layer is excited in mechanical resonance at 689 Hz, and the blue line indicates the resonance behavior of the resonator; (**c**) the measured output spectrum taken from the AlN layer when a low frequency 20 Hz sinusoidal magnetic field is present. The sensor’s PZT layer is excited at 669 Hz, and the blue line indicates the resonance behavior of the resonator; and (**d**) the linearity measurement performed at a magnetic field of 10 Hz, where the noise floor is reached at about 10 nT/Hz^1/2^. This figure is reproduced with permission from AIP Publishing, [90].

**Figure 5 materials-12-02259-f005:**
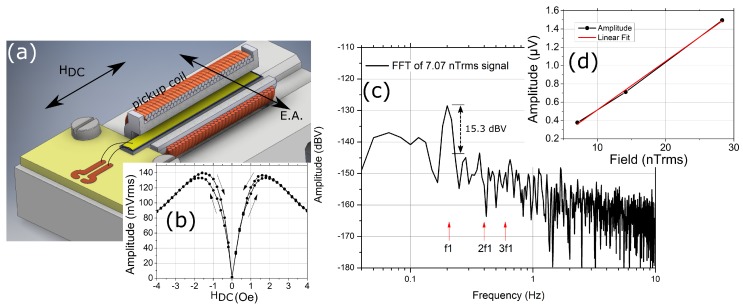
(**a**) Setup showing cantilevered ME composite surrounded by pickup coil (1200 windings). Magnetic easy axis (E.A.) transverse to external field axis (H_DC_); (**b**) coil voltage at mechanical resonance frequency with respect to external magnetic field; (**c**) coil voltage spectrum showing magnetic field of 200 mHz and its margin of about 15 dBV towards noise floor. Absent harmonics at 2f_1_ and 3f_1_; and (**d**) linearity with respect to magnetic signal amplitude. This figure is reproduced with permission from IOP Publishing, [91].

**Figure 6 materials-12-02259-f006:**
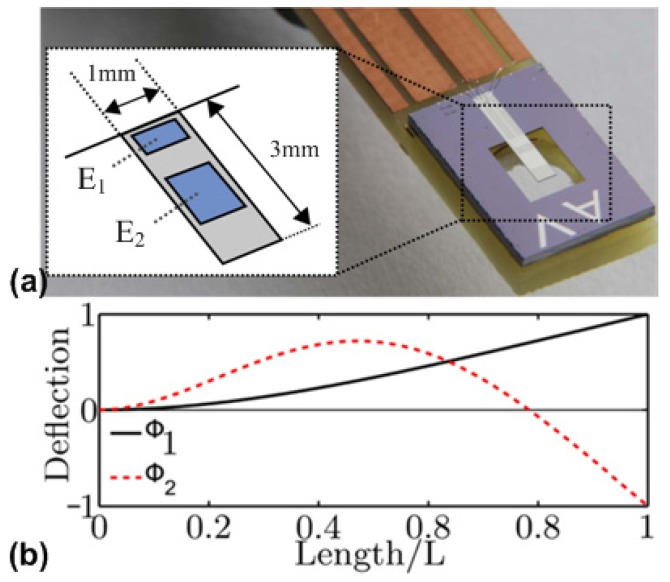
(**a**) Cantilever delta-E effect magnetometer with adapted electrode design, optimized for the second bending mode [107] (© 2016 IEEE); and (**b**) cantilever deflection in the 1st and 2nd transversal bending modes as calculated from Euler-Bernoulli beam theory [106]. This figure is reproduced with permission from AIP Publishing, [106].

**Figure 7 materials-12-02259-f007:**
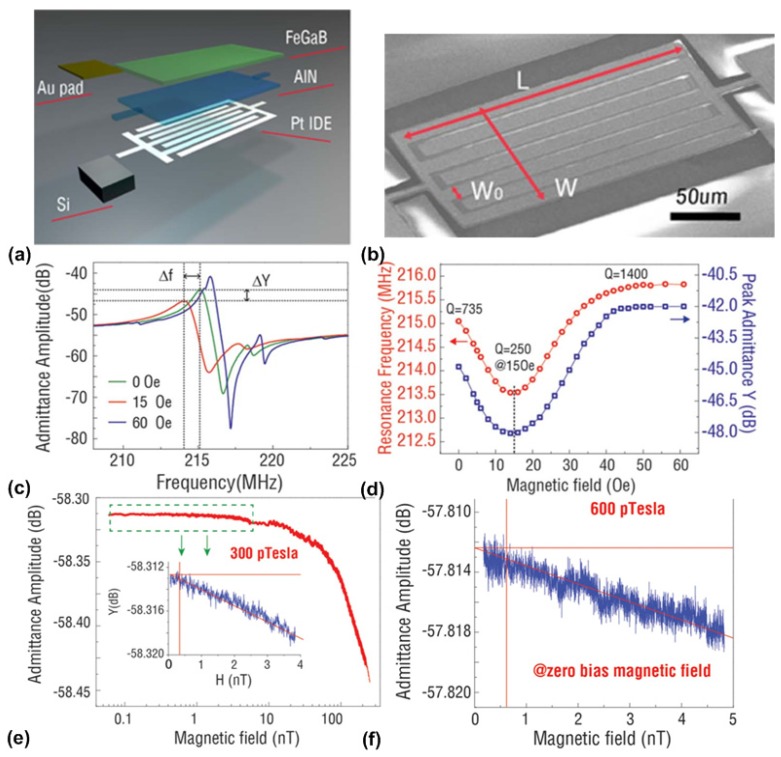
(**a**) Schematic of the layered structure for the nanoplate magnetometer; (**b**) SEM of the fabricated magnetometer; (**c**) measured admittance frequency spectrum of the magnetometer; (**d**) Measured resonant frequency and admittance of the magnetometer; (**e**) measured admittance showing the sensitivity and linearity of magnetometer under DC bias magnetic field of 5 Oe; and (**f**) measured admittance showing the sensitivity and linearity of magnetometer under zero bias magnetic field. This figure is reproduced from [61].

**Figure 8 materials-12-02259-f008:**
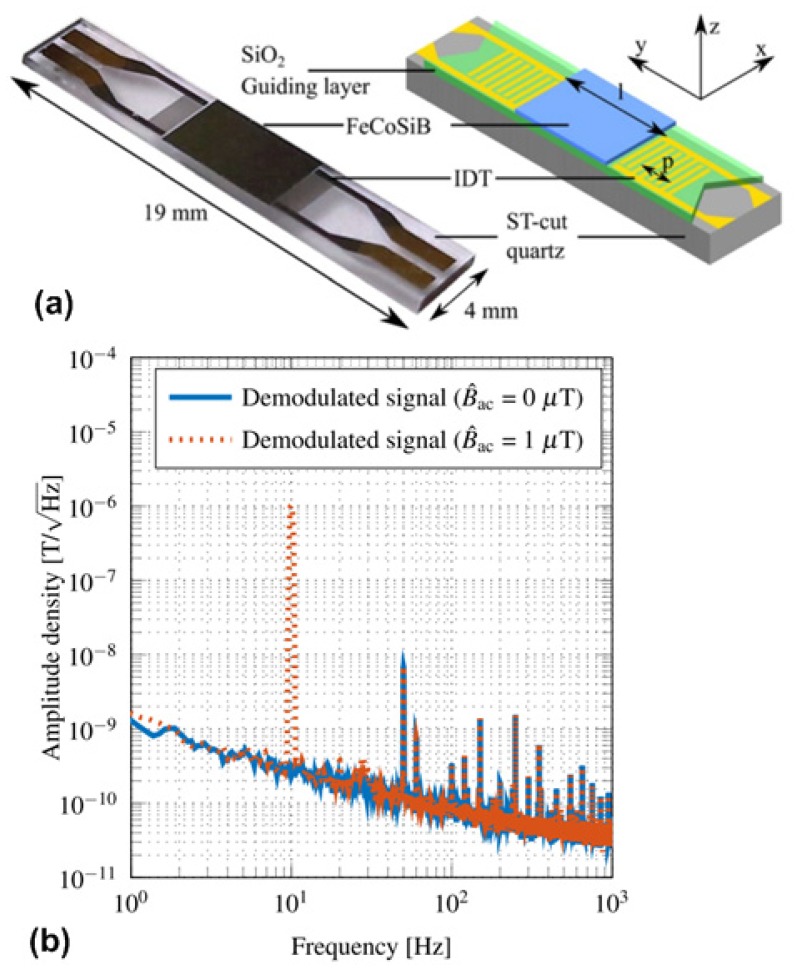
(**a**) A sensor design of the Love wave surface acoustic wave sensor based on ST-cut quartz substrate and a 4.5 μm SiO_2_ guiding layer; and (**b**) equivalent magnetic noise floor of the output signal shows the frequency dependent LOD. An equivalent magnetic noise of 250 pT/Hz^1/2^ at 10 Hz is achieved. This figure is reproduced from [115].

**Figure 9 materials-12-02259-f009:**
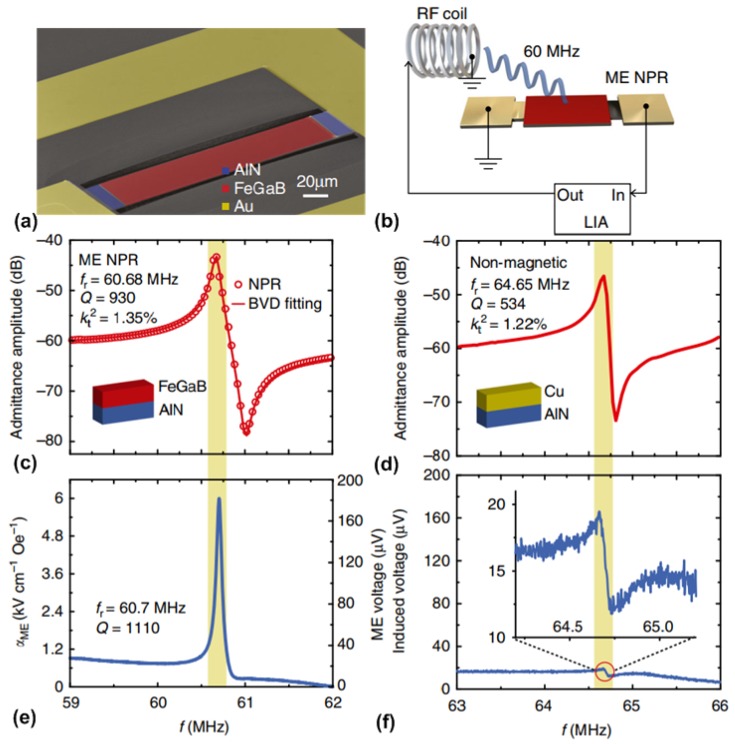
(**a**) SEM of the NPR-type mechanical antenna; (**b**) schematic of the measurement setup using RF coil and high-frequency lock-in amplifier; (**c**) measured admittance frequency spectrum of the NPR-type mechanical antenna; (**d**) measured admittance frequency spectrum of the nonmagnetic Cu/AlN control device; (**e**) ME coupling coefficient (left axis) and the induced ME voltage (right axis) versus the excitation frequency of RF-coil for NPR-type mechanical antenna; and (**f**) measured induced voltage versus the excitation frequency of RF-coil for Cu/AlN control device. The inset illustrates the zoomed-in view of the resonance peak area (red circle). This figure is reproduced from [121].

**Figure 10 materials-12-02259-f010:**
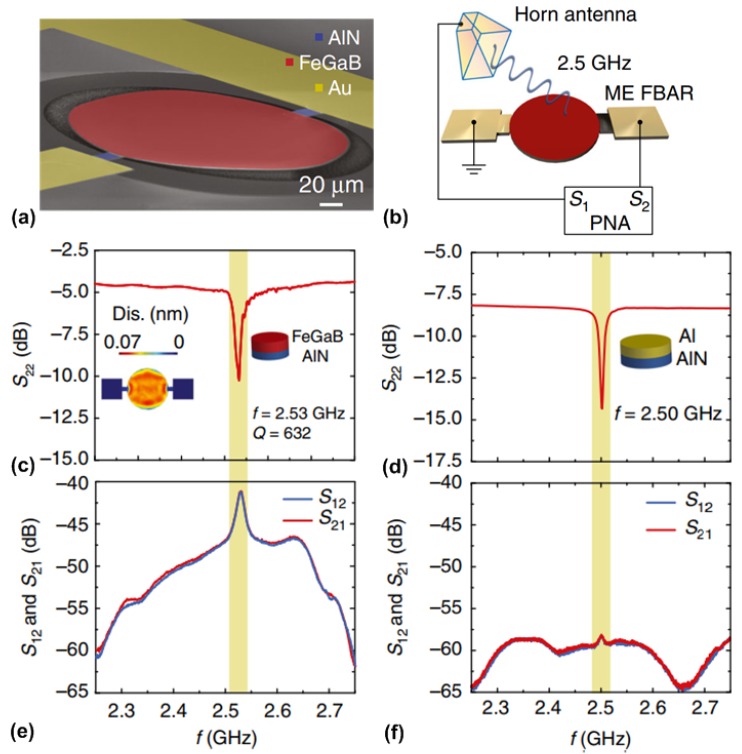
(**a**) SEM of the FABR-type mechanical antenna; (**b**) schematic of the gain measurement setup using Horn antenna and network analyzer; (**c**) measured reflection coefficient (S_22_) of the FABR-type mechanical antenna. The inset shows the out-of-plane displacement of the circular disk at resonant frequency; (**d**) measured reflection coefficient (S_22_) of the nonmagnetic Al/AlN control device; (**e**) Measured transmission coefficients (S_21_ and S_12_) of the FABR-type mechanical antenna; and (**f**) measured transmission coefficients (S_21_ and S_12_) of the nonmagnetic Al/AlN control device. This figure is reproduced from [121].

**Figure 11 materials-12-02259-f011:**
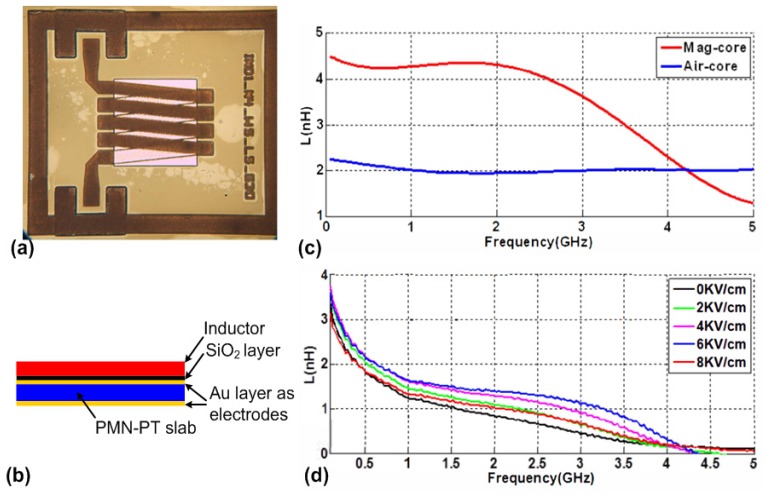
(**a**) Optical image of the tunable inductor; (**b**) schematic of the layered structure of the tunable inductor; (**c**) measured inductances of the inductor with magnetic multilayer films as the inductor core and the one with air core; and (**d**) measured inductance under different E-field applied across the thickness of the PMN-PT slab. This figure is reproduced from [138] and [139] (© 2014 IEEE).

**Figure 12 materials-12-02259-f012:**
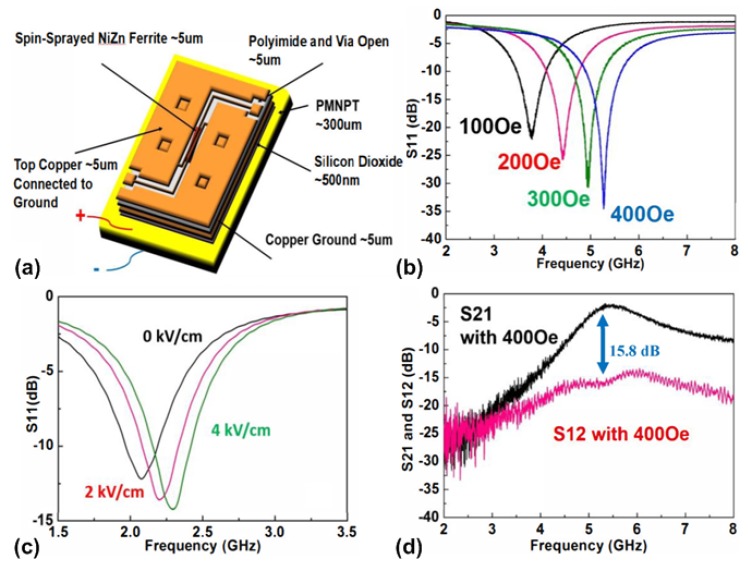
(**a**) Schematic of the tunable filter; (**b**) measured S_11_ of the tunable filter as a function of the DC magnetic field; (**c**) measured S_11_ of the tunable filter as a function of DC electric field across the thickness of the PMN-PT slab; and (**d**) measured S_21_ and S_12_ of the tunable filter at a magnetic bias field of 400 Oe. This figure is reproduced from [151] (© 2015 IEEE).

**Figure 13 materials-12-02259-f013:**
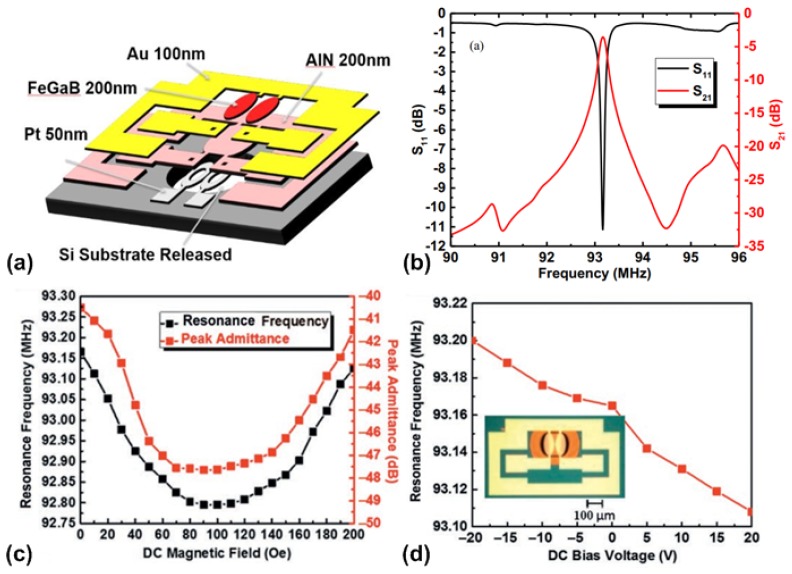
(**a**) Schematic of the tunable filter; (**b**) measured S_11_ and S_21_ of the tunable filter at zero bias field; (**c**) Measured resonant frequency as a function of DC magnetic field; (**d**) measured resonant frequency as a function of DC voltage across the thickness of the AlN film. This figure is reproduced from [153] (© 2016 IEEE).

**Table 1 materials-12-02259-t001:** Summary of the physical mechanisms for various of integrated multiferroic devices.

Device	Type of ME Coupling	Physical Mechanism
Magnetometer	Direct ME coupling	H-field control of electric polarization
Mechanical antenna in transmitting mode	Converse ME coupling	E-field control of magnetization switching
Mechanical antenna in receiving mode	Direct ME coupling	H-field control of electric polarization
Tunable RF inductor	Converse ME coupling	E-field control of permeability
Tunable RF filter	Converse ME coupling	E-field control of magnetostatics surface waves

**Table 2 materials-12-02259-t002:** Summary of characteristics for some of the reviewed integrated multiferroic devices.

Device	Type	Characteristics	Performance	Ref
Magnetometer	Direct detection	(1) Lowest LOD at resonance (2) Narrow bandwidth	LOD = 400 fT/Hz^1/2^ @ 867 Hz	[84]
	Magnetic frequency conversion	(1) Transfer desired low frequency magnetic signal to high frequency via magnetic modulation (2) Wide bandwidth (3) Large power consumption	LOD = 60 pT/Hz^1/2^ @ 10 Hz	[76,89]
	Electric frequency conversion	(1) Transfer desired low frequency magnetic signal to high frequency via electric modulation (2) Wide bandwidth (3) Low power consumption	LOD = 10 nT/Hz^1/2^ @ 10 Hz	[90]
	Delta-E effect	(1) Change of elastic constants upon applied magnetic field (2) Large bandwidth (3) Low power consumption	LOD = 250 pT/Hz^1/2^ @ 10 Hz	[115]
Mechanical antenna	NPR-type	(1) vibrate in lateral direction (2) work at several tens of MHz	α_V_ ^1^ = 6 kV/cm·Oe @ 60.7 MHz	[121]
	FBAR-type	(1) vibrate in thickness direction (2) work at several GHz	G ^2^ = −18 dBi @ 2.5 GHz	[121]
Tunable RF/Microwave devices	Tunable inductor	(1) E-field controlled (2) High Q at GHz	TIR ^3^ = 100% @ 2 ~ 4 GHz	[138]
	Tunable filter	(1) H- or E-filed controlled (2) Nonreciprocity	TFR ^4^ = 39% @ 4 ~ 5 GHz	[151]

^1^ α_V_ denotes the ME voltage coefficient; ^2^ G denotes the measured antenna gain; ^3^ TIR denotes the tunable inductance range; ^4^ TFR denotes the tunable frequency range.

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
