# Peer review of "Mechanical-Resonance-Enhanced Thin-Film Magnetoelectric Heterostructures for Magnetometers, Mechanical Antennas, Tunable RF Inductors, and Filters"

_materials, 2019, doi:10.3390/ma12142259_

Round 1

Reviewer 1 Report

The article reviews the recent progress in application of magnetoelectric thin film heterostructures for special magnetometers, active miniaturized antennas, and tunable RF inductors and filters. The subject is interesting to scientists working in the field of magnetoelectric multiferroics. The review presents the current state of the art and the limitations of application of the mentioned magnetoelectric heterostructures with emphasis on mechanical resonance. A considerable part of the results stems from the groups of the authors in the U.S. and in Germany. The article is clearly and understandably written in very good English.

I have only few minor comments to improve the review:

The title should be more specific to the subject of the work, for example to include "composite", and / or "mechanical resonance", or the often used materials combinations.

The abstract is very long, it should be shortened considerably. It is not neccessary to give here so much content of the review.

In the first pages, the text is often very simple, i.e. basic. For example, other magnetoelectric coupling mechanisms beside the strain coupling are not mentioned here.

Check English in line 697, may be there are other similar errors.

Otherwise, the review is fine for me.

Reviewer 2 Report

See attached file.

Reviewer 3 Report

This manuscript shows a nice review about the most recent progress on several types of integrated multiferroic devices. It would be nice if the authors show a summary table with several type of device because it is sometimes difficult to read this. A suggestion is to make a table with (1) Magnetometer, (2) acoustic antenna, (3) tunable RF/Microwave Devices.

I also suggested to make the abstract more short and attractive to the readers.

Minor suggestions:

Why has ME heterostructure potential capability to integrate with CMOS circuit (line 70-71)

Section 2.1. PZT film is representative material but what is stoichiometric ratio of Zr/Ti? The properties depend on this ratio.

AlN is polycrystalline texture but which texture has PZT ?

Deposition of high quality AlN layer requires high temperature à which deposition technique and which temperature? (line 307-308)

(line 311-312) How is FeCoSiB deposited? And which temperature is AlN layer deposited on FeCoSiB?

(Line 399) What is different between EB-FeCoSiB layer and FeCoSiB layer?

(Line 400) I’m just wonder why the author is talking about “undoped” PZT layer. What about the doped PZT?

Figure 6 show low resolution

(Line 483) 250 nm thick layer in total thickness of 10x multilayer or it is in total 2500 nm multilayer with 10 times of 250 nm layer?

It would be nice if the authors show a summary table with several type acoustic antenna because it is sometimes difficult to read this.

(line 677) which deposition technique? And what is PZN-PT layer?

(line 698) a typo of Al2O3

(line 704) what is PMN-PT layer

(line 745) what is DRIE technique?

Section 5.2, please provide a table with an overview of several materials.
